# Reduction in Serum Magnesium Levels and Renal Function Are Associated with Increased Mortality in Obese COVID-19 Patients

**DOI:** 10.3390/nu14194054

**Published:** 2022-09-29

**Authors:** Patricia Pulido Perez, Jorge Alberto Póndigo de los Angeles, Alonso Perez Peralta, Eloisa Ramirez Mojica, Enrique Torres Rasgado, Maria Elena Hernandez-Hernandez, Jose R. Romero

**Affiliations:** 1Faculty of Medicine, Autonomous University of Puebla, 13 Sur 2901 Col. Volcanes, Puebla 72420, Mexico; 2Internal Medicine Department, University Hospital of Puebla, Mexico. Av 27 Poniente, Los Volcanes, Puebla 72410, Mexico; 3Doctorate in Biological Science, Autonomous University of Tlaxcala, La Loma Xicohtencatl, Tlaxcala 90070, Mexico; 4Division of Endocrinology, Diabetes and Hypertension, Department of Medicine, Brigham and Women’s Hospital, Harvard Medical School, 75 Francis Street, Boston, MA 02115, USA

**Keywords:** SARS-CoV-2, obesity, circulating magnesium levels, glomerular filtration rate

## Abstract

Several studies provide evidence that obesity is a significant risk factor for adverse outcomes in coronavirus disease 2019 (COVID-19). Altered renal function and disturbances in magnesium levels have been reported to play important pathophysiological roles in COVID-19. However, the relationship between obesity, renal function, circulating magnesium levels, and mortality in patients with COVID-19 remains unclear. In this retrospective cohort study, we characterized 390 hospitalized patients with COVID-19 that were categorized according to their body mass index (BMI). Patients were clinically characterized and biochemical parameters, renal function, and electrolyte markers measured upon admission. We found that in patients who died, BMI was associated with reduced estimated glomerular filtration rate (eGFR, Rho: −0.251, *p* = 0.001) and serum magnesium levels (Rho: −0.308, *p* < 0.0001). Multiple linear regression analyses showed that death was significantly associated with obesity (*p* = 0.001). The Cox model for obese patients showed that magnesium levels were associated with increased risk of death (hazard ratio: 0.213, 95% confidence interval: 0.077 to 0.586, *p* = 0.003). Thus, reduced renal function and lower magnesium levels were associated with increased mortality in obese COVID-19 patients. These results suggest that assessment of kidney function, including magnesium levels, may assist in developing effective treatment strategies to reduce mortality among obese COVID-19 patients.

## 1. Introduction

The coronavirus disease 2019 (COVID-19) and the emerging viral variants of severe acute respiratory syndrome coronavirus-2 (SARS-CoV-2) have been a challenge for global health systems with 609M infections resulting in 6.50M deaths worldwide as of 21 September 2022 [1].

Several studies from Europe, Asia, and the USA have independently provided evidence that obesity is a significant risk factor for adverse outcomes in COVID-19 [2]. High body mass index (BMI) has a strong association with increased risk of intensive care admissions, mechanical ventilation, and in-hospital mortality [3,4]. Obesity is a complex metabolic disorder, linked to the development of multiorgan disorders, including hypertension, type 2 diabetes (T2D), cardiovascular diseases (CVD), dyslipidemia, stroke, and certain types of cancer. These disorders may exacerbate and aggravate COVID-19 [5].

Obesity is considered a chronic inflammatory condition that is associated with dysregulated adipose tissue production of pro-inflammatory adipokines, such as the interleukin (IL) family, tumor necrosis factor (TNF) system, monocyte chemoattractant protein-1 (MCP-1), resistin, and leptin [6]. Leptin has been proposed to play an important role in the underlying mechanisms of COVID-19 in obese patients [7]. This is due, in part, to leptin’s control of energy metabolism and immune system function, and its upregulated response to cytokines [8]. Cytokines are key inflammatory molecules that promote thrombotic disorders, immune cell infiltration, and endothelial cell activation, as well as abnormalities associated with bronchial asthma and acute respiratory distress syndrome (ARDS). In addition, cytokines, such as TNF, IL-2, or IL-6, contribute to acute kidney injury and renal failure, as reported in critically ill COVID-19 patients [7,9,10]. Furthermore, conditions linked to obesity, such as endothelial dysfunction and enhanced expression of angiotensin-converting enzyme 2 (ACE2), contribute to acute COVID-19 that confers vulnerability to post-acute sequelae of SARS-CoV-2 infection (Long COVID), an array of signs and symptoms with prolonged multisystem involvement that persist for more than four weeks after an infection has been diagnosed [11,12].

Obese patients, as compared with non-obese subjects, show evidence of magnesium (Mg) deficiency and systemic chronic inflammation, including increases in adipokine release, oxidative stress and increased lymphocyte, neutrophil and macrophage levels [13,14]. It has been proposed that these factors increase the risk of developing severe forms of COVID-19 and may affect susceptibility and the inflammatory responses to SARS-CoV-2, in part, by maintaining and propagating the cytokine release syndrome, a reaction that leads to ARDS, accentuating endothelial dysfunction and coagulopathy. These factors contribute to multi-organ failure and mortality in obese patients [15]. It is well established that deficiency of both Mg and other trace elements can result in abnormal cellular function or damage by increasing or decreasing susceptibility to infections, and negatively affecting the respiratory system [16]. Trace elements are a group of essential metals that include Mg, zinc (Zn), iron (Fe), copper (Cu), and selenium (Se) [17]. Mg is an essential divalent cation required for various metabolic and biochemical functions, with a potential role in the pathogenesis of COVID-19 by contributing to proper immune, vascular, and pulmonary function [18,19].

Mg levels are regulated by its uptake in the gastrointestinal tract, storage in bones by mechanisms regulating electrolyte and water balance in the kidney, including the glomerular filtration [20]. We and others documented that COVID-19 patients show evidence of impaired kidney function that also may contribute to Mg deficiency [21,22,23]. These mechanisms suggest that Mg deficiency and renal disorders contribute to COVID-19 disease severity [23]. However, the relationship between obesity, renal function, and Mg homeostasis in COVID-19 patients remains unclear.

In this study, we explored the hypothesis that obese patients with COVID-19, as compared to non-obese, show impaired renal function and disordered circulating Mg levels that are associated with COVID-19-related mortality. To test our hypothesis, we characterized renal function and circulating Mg levels in a cohort of COVID-19 patients admitted to the University Hospital of Puebla in the City of Puebla in Mexico.

## 2. Materials and Methods

### 2.1. Data Source, Subjects, and Setting

This is a retrospective, cohort study that included 390 adult patients who were diagnosed with SARS-CoV-2 and hospitalized in the University Hospital of Puebla (HUP), in the city of Puebla in Mexico, between 24 March 2020–22 March 2022. The study was approved by the Institutional Ethics Committee of the HUP (Registry CEIHUP 2020/033). This study was conducted using data collected from routine clinical practice and waived the requirements for informed consent. Diagnosis was by real-time PCR testing of a nasopharyngeal sample and validated by National Institute of Epidemiologic Diagnosis and Reference (INDRE) in Mexico. Our study follows the recommendations for strengthening of reports of observational studies, as noted in Strengthening the Reporting of Observational Studies in Epidemiology (STROBE) guidelines [24].

Demographic and clinical characteristics, symptoms, comorbidities, and laboratory parameters were extracted from hospital electronic health records. Clinical information collected included medical and exposure history, comorbidities, vital signs (cardiac and respiratory rate, blood pressure), symptoms, treatment regimens for COVID-19, and other comorbidities. Additional data collected were respiratory support (e.g., nasal tube, non-invasive, and invasive mechanical ventilation); oxygen saturation (SatO_2_); partial pressure of oxygen/fraction of inspired oxygen (PaO_2_/FiO_2_), an indirect measure of the lung injury [25]; organ failure as estimated by the Sequential Organ Failure Assessment (SOFA) score [26]; length of stay and prognosis (home discharge or death). Laboratory evaluations included the determination of metabolic status: plasma glucose (PG), uric acid, cholesterol; blood cell counts: erythrocytes, leucocytes, lymphocytes, neutrophils, and platelets; coagulation; and inflammatory markers: D-dimer, high sensitive C-reactive protein (hs-CRP), and procalcitonin levels. Laboratory evaluations were repeated as needed during hospitalization. In this report, we show data collected at hospital admission and the last measurement prior to home discharge or death.

### 2.2. BMI Categorization

BMI was calculated as weight/height^2^ (kg/m^2^), based on the height and weight measured at hospital admission. We categorized patients according to their BMI using the WHO adult criteria: normal weight was defined as BMI between 18.5 and 24.9 kg/m^2^, a BMI between 25 kg/m^2^ and 29.9 kg/m^2^ was considered overweight and a BMI of 30 kg/m^2^ or higher was considered obese [27].

### 2.3. Evaluation of Renal Function

Renal function was assessed by the estimated glomerular filtration rate (eGFR) calculated according to the Chronic Kidney Disease Epidemiology Collaboration (CKD-EPI) equation [CKD-EPI= 141 × min (SCreatinine/κ,1)^α^ × max (SCreatinine/κ,1)^−1.209 ×^ 0.993^Age(years)^ (X 1.018 if female), where k is 0.7 for females and 0.9 for males, α is −0.329 for females and −0.411 for males] [28]. As previously described, values of eGFR ≥ 90 mL/min per 1.73 m^2^ were consider normal function, an eGFR of 89 to 60 mL/min per 1.73 m^2^ indicated mild decreased function, and an eGFR level less than 60 mL/min per 1.73 m^2^ was considered as a marker of disordered renal function that represents loss of half or more of the adult level of normal kidney function [29]. Creatinine, urea, blood urea nitrogen (BUN) and circulating electrolyte status (Mg, sodium, potassium, calcium, chloride, and phosphorus) were also evaluated.

Determination for sodium, potassium, and chloride levels was carried out using appropriate ion-selective electrodes. Serum calcium levels has been measured using the ortho-Cresolphthalein dye method and phosphomolybdate colorimetric test for phosphorus levels. Serum Mg levels were determined by colorimetric assay using the Xylidyl blue method. Normal range of serum Mg was defined as 1.5–1.9 mEq/L (0.75 mmol/L–0.95 mmol/L), hypomagnesemia as a serum Mg concentration lower than 1.5 mEq/L (<0.75 mmol/L), and hypermagnesemia as a serum Mg concentration greater than 1.9 mEq/L (>0.95 mmol/L) [30].

### 2.4. Statistical Analyses

Categorical variables were expressed as percentages and compared by Chi-square test. Continuous variables were expressed as means ± standard deviations or median and interquartile range (IQR), from 25th to 75th percentiles. The Kolmogorov–Smirnov test was used to determine the normality of the data distribution. The association between parameters was determined by Spearman’s correlation coefficient. The comparison between admission vs. home discharge or death were analyzed by Wilcoxon signed-rank test. Cox regression proportional hazards model with cumulative risk adjusted for age and biological sex was used for survival analysis, considering renal function and electrolyte status in obese patients with COVID-19. Several covariates were selected for analysis as potential confounding variables in our regression analyses, these included age, sex, comorbidities, disease severity, electrolyte levels, D-dimer, hs-CRP, and lymphocyte count. Statistical analyses were performed using the Statistical Package for the Social Sciences program for Windows version 25.0 (SPSS, Chicago, IL, USA). Statistical charts were generated using GraphPad Prism for Windows version 8.0.1 (San Diego, CA, USA). For all the statistical analyses, *p* < 0.05 was considered significant.

## 3. Results

Our study included a total of 390 patients. Among these, 238 were men and 152 were women, between the ages of 27 and 99 years. Characteristics of the entire cohort are summarized in Table 1. In our cohort 239 (61.3%) patients were discharged home and 151 (38.7%) patients died within 11.5 ± 7.9 days after hospital admission. Table 2 shows the comparison between admission and discharge home or death. In the entire cohort, the patients who died in hospital were older than those who were discharged home (68.1 ± 12.3 vs. 61.73 ± 13.7 years, *p* < 0.001). Patients that died had increased inflammatory and pro-thrombotic status (high levels of neutrophils, leucocytes, D-dimer, hs-CRP, and procalcitonin), as compared with home discharged patients (Table 2).

The classification, according to BMI, shows that 24.4% (95 patients) had normal weight, whereas 41.0% (160 patients) were overweight and 34.6% were obese (Table 1). Obese patients were younger (61.7 ± 14.2 years, *p* = 0.001) than overweight and normal weight patients (64.7 ± 13.3 and 67.8 ± 12.5 years, respectively). BMI correlated with COVID-19 associated mortality (*p* = 0.001). Obese patients had more frequency of fatal outcome than those with normal weight (45.2% obese patients died, 40.0% overweight patients died, and 27.4% normal weight patients died). Analyses of the comorbidities in our cohort show that the proportion of hypertension in the group of normal weight patients was 47.4%, 50.6% for overweight, and 55.6% for obese patients. Comparison between deceased vs. discharged home group shows differences only among the obese group (66.1% vs. 47.4%, *p* = 0.030). Diagnosis of diabetes in the group of obese patients was 48.1%, which is not significantly different for those with normal weight (49.5%, *p* = 0.844). In a similar manner, the comparison between deceased vs. discharged home obese patients did not show significant differences (47.4% vs. 49.2%, *p* = 0.838). The complications associated with death included acute respiratory distress syndrome (ARDS; 260, 66.6%), pneumonia (111, 28.4%), and septic shock (102, 26.1%). In our cohort, we found that 240 patients (61.5%) had renal alterations; 103 (26.4%) subjects had mildly decreased kidney function, defined by eGFR as 89 to 60 mL/min per 1.73 m^2^; and 137 (35.1%) patients had disordered kidney function, defined by eGRF <60 mL/min per 1.73 m^2^. Analyses of renal function by BMI distribution showed an increase in eGFR at discharge (home discharge or death), except in obese patients who died, who had a significant reduction in eGFR [52.0 (33.8–96.2 to 29.5 (19.4–47.6) mL/min per 1.73 m^2^, *p* < 0.0001], Figure 1a,b.

The median serum Mg concentration of our total population was 2.0 (1.80–2.21) mEq/L. Hypomagnesemia (defined as a serum Mg levels < 1.5 mEq/L) was observed in 13 (3.3%) patients, 106 (27.2%) had “normal” serum Mg levels (1.5–1.9 mEq/L), whereas 271 (69.5%) patients had hypermagnesemia (Mg levels > 1.9 mEq/L). Figure 1c,d shows Mg levels by BMI distribution in overweight and obese patients showed a reduction in Mg levels between admission and the last measurement prior to death [2.03 (1.89–2.21) to 1.96 (1.79–2.17)] mL/min per 1.73 m^2^, *p* = 0.039 for overweight patients and [2.10 (1.90–2.44) to 1.79 (1.62–2.00)] mL/min per 1.73 m^2^, *p* < 0.0001 for obese patients).

Correlation analyses of Mg levels and markers of renal function showed that Mg levels in overweight patients that died were associated with potassium (Rho: 0.310, *p* = 0.014) and eGFR (Rho: −0.257, *p* = 0.040), whereas Mg levels correlated with potassium (Rho: 0.277, *p* = 0.034), urea (Rho: 0.378, *p* = 0.003), and eGFR (Rho: −0.298, *p* = 0.022) in obese patients that died (Figure 2). No significant correlation was observed for Mg levels and renal markers in home discharged patients. In addition, Mg levels in overweight patients that died were associated with organ failure, as measured by SOFA score (Rho: 0.342, *p* = 0.006), CT score (Rho: 287, *p* = 0.021), and hs-CRP (Rho: 0.421, *p* = 0.012). In obese patients that died, Mg levels were associated with lower respiratory rate (Rho: −0.257, *p* = 0.049), cardiac rate (Rho: −0.319, *p* = 0.015), and SatO_2_ (Rho: −0.351, *p* = 0.006).

Cox regression proportional hazards model with cumulative risk was used for survival analyses, including renal function and electrolyte status, in obese patients. Considering the importance of the age in the comparison groups, we ran survival analyzes and Cox regression adjusted by age and other biological differences, including biological sex (Figure 3). Our results show significant association between eGFR and Mg levels (HR 0.213, 95% CI 0.077 to 0.586, *p* = 0.003) and creatinine (HR 0.515, 95% CI 0.284 to 0.934, *p* = 0.029). No significant correlation was observed for Mg levels and renal markers in home overweight death or patients discharged home (Table 3).

## 4. Discussion

In this study, we assessed the relationship between obesity, renal function, circulating Mg levels, and mortality in Mexican patients hospitalized for COVID-19. Our novel results show that reduced renal function and low serum Mg levels were associated with increased mortality in obese COVID-19 patients. To the best of our knowledge, there are no reports examining BMI and Mg status in COVID-19 patients.

Our findings extend reports of others, showing that Mg deficiency is commonly reported in obese patients and associated with kidney disease and has been proposed to play a role in the pathophysiology of COVID-19 [22,31,32]. The mechanisms to explain our findings are unknown. There is evidence that not only is Mg deficiency a risk factor for development of insulin resistance and diabetes but there also is evidence that diabetes contributes to Mg deficiency [33,34]. Lower magnesium levels in deceased obese patients may be due to the high contribution in mortality among patients with diabetes. Of importance, in our cohort, obese COVID-19 patients that did not improve their kidney function while hospitalized, maintained low circulating Mg levels, and were particularly susceptible to poor survival rates, when compared to normal weight or overweight patients.

In our cohort, we observed decreased Mg levels in obese patients who died. These values are within the normal range. It is important to note that there is evidence that low Mg levels stimulate a compensatory increase in Mg reabsorption and promotes paracellular magnesium transport by the kidney, maintaining serum magnesium levels within the “normal” range, even when there is a severe reduction in GFR (<30 mL/min mL/min per 1.73 m^2^) [21,35]. This effect is not apparent with the other electrolytes, whose levels are affected regardless of the outcome of COVID-19 (home discharged or death) or in obese patients.

Low Mg levels have been reported to stimulate development of the cytokine storm, a reaction that leads to ARDS, accentuates pro-thrombotic status, and promotes multiple organ failure [15,36]. In our cohort, Mg levels were associated with COVID-19 complications, including ARDS, septic shock, and disseminated intravascular coagulation. In addition, patients that died had lower Mg levels and increased inflammatory and pro-thrombotic status (high levels of D-dimer, hs-CRP, and procalcitonin), as compared with patients that were discharged home.

We found that high BMI was associated with COVID-19 mortality. Our results confirm and extend previous findings, showing that COVID-19 is more deadly in people with obesity [37]. Several biological mechanisms by which COVID-19 can affect people with obesity have been proposed but no clear consensus exits [38]. Obesity is characterized by an excess of adipose tissue that is linked to impaired immune function, chronic inflammation, pro-thrombotic status, and developing the cytokine storm [11,39]. Moreover, obesity has been proposed as an important contributor to development of Long COVID-19 [12]. In addition, there is evidence to suggest that adipose tissue can act as a viral reservoir for viruses, including SARS-CoV and type A influenza, and as such may contribute to prolonged viral shedding [36,40].

Obesity is also a potent risk factor for the development of kidney disease [41]. Our data are consistent with the growing body of information that kidney disease is a common complication among hospitalized COVID-19 patients and an important risk factor for COVID-19-related mortality [42,43,44]. In our cohort, we observed that more than 60.0% of our patients had renal dysfunction: 26.4% had mildly decreased kidney function (eGFR of 89 to 60 mL/min per 1.73 m^2^) and 35.1% had disordered kidney function (eGRF < 60 mL/min per 1.73 m^2^). These results suggest that renal dysfunction in obese patients is closely related to disease progression to mortality through unclear mechanisms.

Adipose tissue ACE2 expression levels correlate with increased COVID-19 associated cardio-metabolic risk factors [45,46]. In addition, increased serum ACE2 levels have been reported in obese patients, suggesting an obesity-induced regulation of ACE2 [47,48]. ACE2 functions as a SARS-CoV-2 receptor and is a key determinant of SARS-CoV-2 infectivity in the host [49]. ACE2 is widely expressed in the respiratory system and other tissues in the body, including kidney and adipose tissue [45,50], and as such may contribute to multiorgan and kidney tropism, affecting renal function and worsening local inflammatory responses [51,52]. However, the relationship between the ACE2 expression in adipose and kidney tissue in the context of kidney damage in obese COVID-19 patients remains unexplored.

Our study has some limitations. This is a retrospective survey that does not allow us to provide for cause and effect. However, we document simultaneous measurements of renal function and Mg levels in hospitalized COVID-19 patients that have important clinical relevance. Moreover, we cannot rule out that the prevalence of renal alterations at admission in our cohort may be the result of undiagnosed renal history amongst our population and, as such, may confound our analyses. Consequently, larger prospective studies are required to confirm our results. Our study provides a rationale for developing such trials.

## 5. Conclusions

Reduced renal function and Mg levels were associated with increased mortality in obese COVID-19 patients. These results suggest that assessment of kidney function, including Mg levels, may assist in developing effective treatment strategies to reduce mortality among obese COVID-19 patients.

## Figures and Tables

**Figure 1 nutrients-14-04054-f001:**
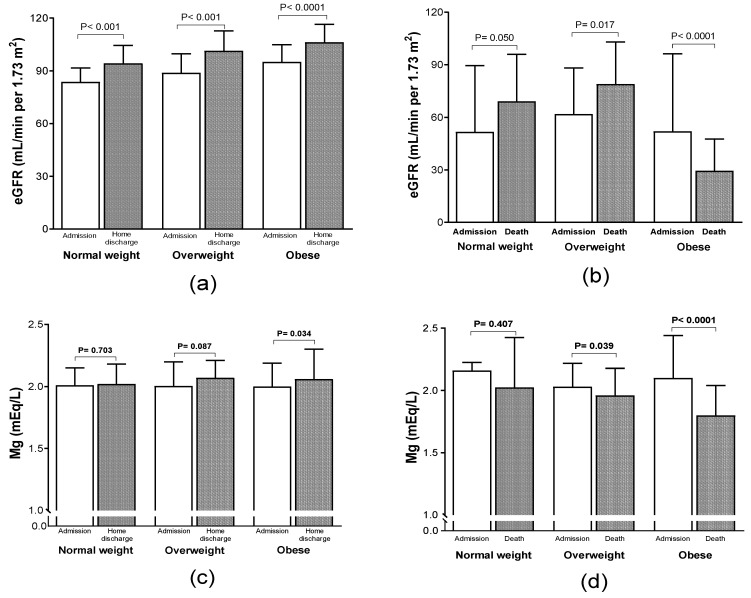
Renal function and magnesium levels by BMI distribution in COVID-19 patients. (**a**) renal function in home discharge patients; (**b**) Renal function in death patients; (**c**) Magnesium levels in home discharge patients; (**d**) Magnesium levels in death patients. Data are presented as median and interquartile range. The comparison between the groups were analyzed by Wilcoxon signed-rank test. *p* < 0.05 was considered significant.

**Figure 2 nutrients-14-04054-f002:**
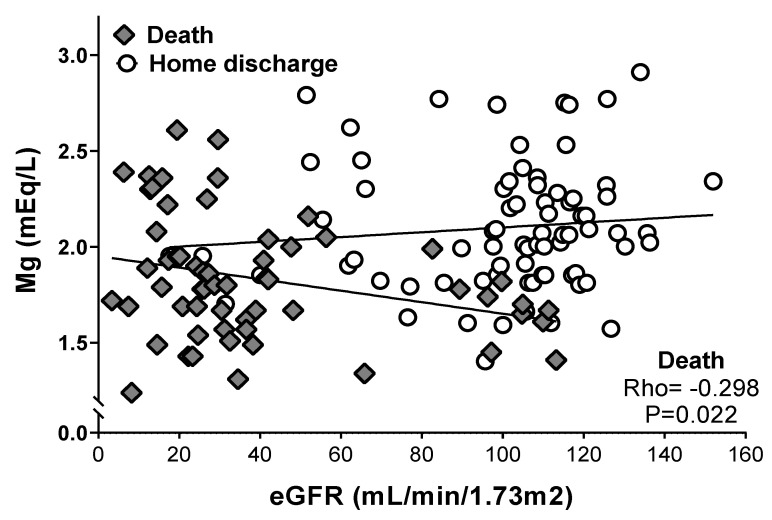
Magnesium levels correlate with estimated glomerular filtration rate (eGFR) in obese COVID-19 death patients versus obese COVID-19 home discharge patients. The association between parameters was determined by Spearman’s correlation coefficient. *p* < 0.05 was considered significant.

**Figure 3 nutrients-14-04054-f003:**
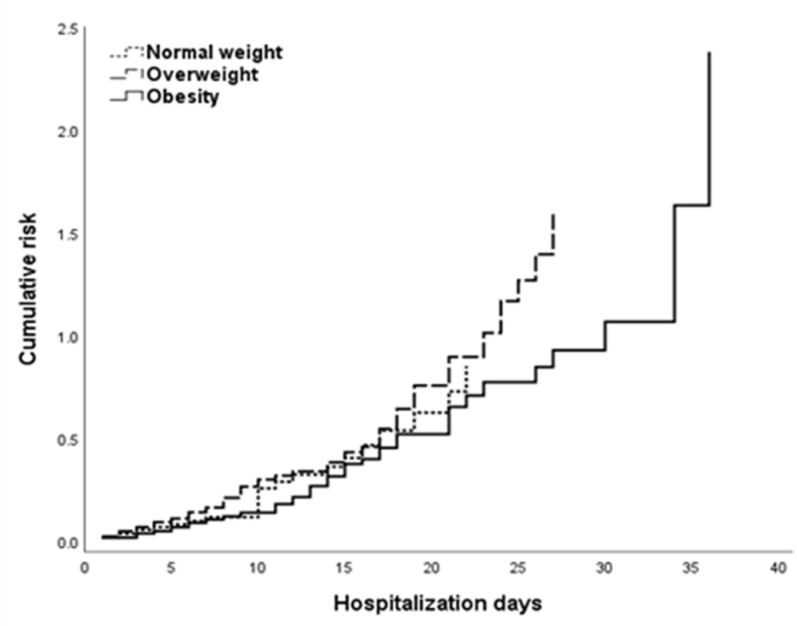
Survival analysis from Cox regression proportional hazards model by BMI distribution in COVID-19 patients.

**Table 1 nutrients-14-04054-t001:** Summary of the characteristics of the entire cohort of COVID-19 patients, (*n* = 390).

Demographic Characteristics	Admission
Age, yrs	64.4 ± 13.6
Sex male, *n* (%)	238 (61.0)
Clinical characteristics	
BMI, Kg/m^2^	28.4 ± 5.3
SARS vaccine, *n* (%)	97 (24.9)
Respiratory rate, breaths/min	23.7 ± 6.0
Cardiac rate, beats/min	83.7 ± 17.3
Systolic blood pressure, mm Hg	123.1 ± 20.0
Dyastolic blood pressure, mm Hg	71.7 ± 11.5
Hospitalization days	9.9 ± 8.4
SatO_2_, %	86.1 ± 11.5
PaO_2_/FiO_2_ ratio	230.4 ± 128.5
CT score	14.6 ± 5.2
SOFA	3.2 ± 1.7
Intubation, *n* (%)	120 (30.8)
Death, *n* (%)	151 (38.7)
Symptoms, *n* (%)	
Dyspnea	325 (83.3)
Myalgia	260 (66.7)
Cough	256 (65.6)
Arthralgia	228 (58.5)
Fatigue	215 (55.1)
Fever	189 (48.5)
Headache	158 (40.5)
Odynophagia	98 (25.1)
Diarrhea	57 (14.6)
Comorbidities, *n* (%)	
Hypertension	201 (51.5)
Diabetes	176 (45.1)
Smoking	74 (19.0)
Kidney chronic disease	35 (9.0)
Autoimmune disease	16 (4.1)
COVID-19 treatment, *n* (%)	
Anticoagulant	369 (94.6)
Glucocorticoids	329 (84.3)
Azithromycin	57 (14.6)
Hydroxychloroquine	27 (6.9)
Oseltamivir	14 (3.5)

Data are shown as means ± standard deviation or median and interquartile range. Abbreviations: body mass index (BMI); oxygen saturation (SatO_2_); partial pressure of oxygen/fraction of inspired oxygen (PaO_2_/FiO_2_); sequential organ failure assessment (SOFA).

**Table 2 nutrients-14-04054-t002:** Comparisons of the laboratory measures, renal function, and electrolyte status between admission, home discharge, and death (*n* = 390).

Indicator	Admission	Home Discharge	Death	*p*-Value
Laboratory				
Glucose, mg/dL	133.5 (109.7–186.5)	112.0 (95.0–137.0) ^§^	130.0 (104.7–167.7) ^#^	<0.0001
Uric acid, mg/dL	5.0 (3.4–7.0)	3.6 (2.7–5.3)	4.9 (2.0–7.4)	0.170
Cholesterol, mg/dL	128.5 (106.0–155.2)	141.0 (117.0–166.0) ^§^	129.0 (109.5–150.5) ^#^	0.004
Erythrocytes, × 106/μL	4.7 (4.3–5.1)	4.7 (4.3–5.1) ^§^	4.3 (3.7–4.8) ^#^	0.002
Neutrophils, × 103/μL	7.6 (5.5–10.9)	6.0 (4.3–7.8) ^§^	10.7 (6.6–14.9) ^#^	<0.0001
Leucocytes, × 103/μL	8.5 (6.8–12.3)	7.4 (5.4–9.3) ^§^	12.4 (8.4–16.1) ^#^	<0.0001
Lymphocytes, × 103/μL	0.6 (0.4–1.0)	0.8 (0.5–1.2) ^§^	0.6 (0.3–0.8)	0.003
Platelets, × 103/μL	239.0 (176.2–308.7)	282.0 (195.0–365.0) ^§^	200.0 (114.0–277.0) ^#^	<0.0001
D-dimer, ng/mL	402.0 (242.0–701.0)	468.0 (287.5–889.2)	1637.0 (481.0–2651.0)^#^	<0.0001
hs-CRP, mg/dL	10.2 (3.1–19.7)	6.7 (1.9–12.8) ^§^	19.5 (10.5–24.6) ^#^	<0.0001
Procalcitonin, ng/mL	0.23 (0.10–0.76)	0.15 (0.08–0.44)	0.40 (0.15–1.51) ^#^	<0.0001
Renal function				
eGFR (mL/min/1.73m^2^)	80.9 (45.7–98.1)	100.2 (86.4–112.6) ^§^	42.0 (24.1–90.1)	<0.0001
Creatinine (mg/dL)	0.9 (0.7–1.3)	0.6 (0.5–0.8) ^§^	1.2 (0.6–2.3)	<0.0001
Urea (mg/dL)	42.8 (29.9–71.7)	40.6 (27.8–53.5)	70.6 (41.7–116.2) ^#^	<0.0001
BUN (mg/dL)	20.0 (15.0–34.0)	19.0 (13.0–27.0)	30.0 (19.0–51.0) ^#^	<0.0001
Electrolytic status				
Magnesium (mEq/L)	2.02 (1.85–2.21)	2.05 (1.85–2.21) ^§^	1.90 (1.70–2.12) ^#^	<0.0001
Sodium (mEq/L)	137.0 (134.0–140.0)	139.0 (136.0–140.0) ^§^	141.0 (138.0–144.0) ^#^	<0.0001
Potassium (mEq/L)	4.2 (3.9–4.6)	4.2 (3.9–4.5)	4.6 (4.0–5.3) ^#^	<0.0001
Calcium (mEq/L)	8.6 (8.2–9.0)	8.4 (8.1–8.8) ^§^	8.1 (7.8–8.6) ^#^	<0.0001
Chloride (mEq/L)	103.0 (100.0–107.0)	105.0 (102.0–108.0) ^§^	107.0 (101.0–109.0) ^#^	0.061
Phosphorus (mEq/L)	3.3 (2.8–3.9)	3.4 (2.9–4.0) ^§^	4.0 (3.3–5.6) ^#^	<0.0001

Data are presented as median and interquartile range. *p*-value in the table correspond to the comparison between the home discharge vs. death COVID-19 patients, analyzed by Kruskal–Wallis test. The comparison between admission vs. home discharge or death were analyzed by Wilcoxon signed-rank test. ^§^ Admission vs. home discharge; ^#^ admission vs. death. *p* < 0.05 was considered significant. Abbreviations: high sensitivity C-reactive protein (hs-CRP); estimated glomerular filtration rate (eGFR); and blood urea nitrogen (BUN).

**Table 3 nutrients-14-04054-t003:** Cox regression proportional hazards model with cumulative risk considering renal function and electrolyte status in obese patients with COVID-19. The analysis was adjusted for age and biological sex.

Parameter	HR	95.0% CI		*p* Value
eGFR (mL/min/1.73 m^2^)	0.966	0.948	0.984	0.000
Creatinine (mg/dL)	0.515	0.284	0.934	0.029
Urea (mg/dL)	1.054	0.852	1.303	0.630
BUN (mg/dL)	0.887	0.563	1.397	0.605
Magnesium (mEq/L)	0.213	0.077	0.586	0.003
Sodium (mEq/L)	1.023	0.962	1.088	0.468
Potassium (mEq/L)	1.391	0.901	2.146	0.136
Calcium (mEq/L)	1.012	0.628	1.630	0.962
Clorom (mEq/L)	1.020	0.958	1.086	0.537
Phosphorus (mEq/L)	1.130	0.952	1.340	0.161

## Data Availability

The data that support the findings of this study are available upon request from the corresponding author following reasonable request and will be considered on a case-by-case basis from qualified researchers with approvals from the Ethics Committee, Institutional review board, and executed institutional data transfer agreements.

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
