# Peer review of "Reduction in Serum Magnesium Levels and Renal Function Are Associated with Increased Mortality in Obese COVID-19 Patients"

_nutrients, 2022, doi:10.3390/nu14194054_

Round 1

Reviewer 1 Report

Very nice and interesting study -data are well presented amd evaluated -  the conclusions are oK (neither pessi-nor too optimistic)

Reviewer 2 Report

1. In the Introduction, it is advisable to give more details about the multi-organ disorders associated with obesity and aggravating COVID-19 disease. For example, that obesity is a major factor in the development of arterial hypertension, dyslipidemia, type 2 diabetes mellitus, cardiovascular disease, bronchial asthma, musculoskeletal diseases. In addition, to briefly characterize the effects of adipokines, in particular leptin, which can stimulate cellular immune response, influence the production of pro-inflammatory cytokines, cause renal failure, the development of atherosclerosis and bronchial asthma. That is, to strengthen the arguments of greater vulnerability of obese patients to COVID-19, including at the expense of possible renal failure, than patients without obesity.

Also in the introduction, it is useful to introduce information about the trace element status of COVID-19 patients, which includes Mg and other elements, in order to make sure of the special role of Mg, the reduction of which can contribute to the mortality of obese patients (See, for example, Li Y, Luo W and Liang B (2022) Circulating trace elements status in COVID-19 disease: A meta-analysis. Front. Nutr. 9:982032. doi: 10.3389/fnut.2022.982032).

2. The article contains some conflicting information about the age of those who died and were discharged home and the age of obese people whose mortality was higher than that of non-obese patients: «patients who died in the hospital were older than those who were discharged home (68.1 ± 12.3 vs. 61.73 ± 13.7 144 years, P<0.001) (line 143).

But these data are at odds with those given in lines 163, 164: Obese patients were younger (61.7 ± 14.2 years, P=0.001), than overweight and normal weight patients (64.7 ± 13.3 and 67.8 ± 12.5 years, respectively). Moreover, Body Mass Index (BMI) correlated with COVID-19 associated mortality (P=0.001).

This contradiction could have been resolved, if the authors had introduced the age distribution in the comparison groups. It is possible that there may be a bi-modal age distribution in the group of the dead with obesity, including the old group and the more younger one.

3. The title of Table 1 repeats the name of the indicators given in the table to characterize the cohort COVID-19 patients. It is better to give a generalized name. For example, "Statistical values of the indicators (means ± standard deviation), characterizing of the entire cohort of COVID-19 patients.

In addition, in Table 1 BMI, Kg/m2, weight and obesity refer to patients' somato-metric characteristics rather than demographic characteristics.

4. It would also be better to change the name of Table 2. For example, "Values of indicators (median and interquartile range), characterizing COVID-19 patients (n=390). In addition, Table 2 would be useful to include a column with the reference values of these indicators

5. The representation of the results in Figure 1 contains redundant information in the form of a set of individual indicators plotted on the columns of data.The figure would look more concise and would not lose in informativeness, if it was represented by only values with «median and interquartile range».

6. In line 194, the letters (b-c) in Figure 1 should be changed to (c-d).

7. Can one attribute a special role to magnesium in mortality of obese patients, based on the decrease in its level prior to death, relative to the level at admission (from 2.1 ± 0.4 to 1.8 ± 0.3 mL/min per 1.73 m2, P<0.0001, correspondingly), if these values are in the reference range (1.5–1.9 mEq/L)? Hypomagnesemia (defined as a serum Mg levels <1.5 mEq/L) was only observed in 13 (3.3%) patients, 106 (27.2%) had “normal” serum Mg levels (1.5–1.9 mEq/L), whereas 271 (69.5%) patients had hypermagnesemia (Mg levels >1.9 mEq/L) (Lines 192-194).

It may be that lower magnesium levels in deceased obese patients are due to the high contribution in mortality patients with diabetes. There is evidence that not only is Mg deficiency a risk factor for the development of insulin resistance and diabetes mellitus, but also diabetes mellitus is a common reason for Mg deficiency (B. von Ehrlich et al: Significance of magnesium in insulin resistance, metabolic syndrome, and diabetes - recommendations of the Association of Magnesium Research e.V. //Trace Elements and Electrolytes, January 2017. 34(07). DOI 10.5414/TEX01473).

Indirect evidence of this possibility comes from data on comorbidity (Table 1), where 45.1% of COVID-19 patients had diabetes. If the authors are able to estimate percentage of diabetic patients among deceased obese patients, this could contribute to understanding the causal relationships between comorbidity, renal failure, magnesium levels, and COVID-19 mortality in obese patients.
